# Human–Animal Interactions in Zoos: What Can Compassionate Conservation, Conservation Welfare and Duty of Care Tell Us about the Ethics of Interacting, and Avoiding Unintended Consequences?

**DOI:** 10.3390/ani10112037

**Published:** 2020-11-04

**Authors:** Mark James Learmonth

**Affiliations:** Animal Welfare Science Centre, The University of Melbourne, Parkville, Victoria 3010, Australia; mlearmonth@student.unimelb.edu.au

**Keywords:** human–animal interactions, animal–visitor interactions, compassionate conservation, conservation welfare, duty of care, animal ethics, zoo animals

## Abstract

**Simple Summary:**

This article is an examination of human–animal interactions in zoos from an ethical perspective, their benefits to both human and animal participants, and also their potential risks and ethical dilemmas. Contact with animals can be beneficial for all parties involved, and can indeed lead to pro-conservation and respect for nature behaviours being adopted by humans after so-called “profound experiences” of connecting or interacting with animals. Yet, human–animal interactions may also increase certain individuals’ desires for inappropriate wild-animal ‘pet’ ownership, and can convey a false sense of acceptability of exploiting animals for “cheap titillation”. Three ethical frameworks that may be beneficial for ethically run zoos to incorporate when considering human–animal interactions are: Compassionate Conservation, Conservation Welfare and Duty of Care. Human–animal interactions in zoos may be acceptable in many circumstances, and may be beneficial to both animal and human participants; however, they must be closely monitored through welfare tracking tools. Melding Duty of Care and the two Conservation ethical frameworks would be ideal for assessing the ethical acceptability of such interactions.

**Abstract:**

Human–animal interactions (HAIs) in zoos can be rewarding for both humans and animals, but can also be fraught with ethical and welfare perils. Contact with animals can be beneficial for all parties involved, and can indeed lead to pro-conservation and respect for nature behaviours being adopted by humans after so-called “profound experiences” of connecting or interacting with animals. Yet, human–animal interactions may also increase certain individuals’ desires for inappropriate wild-animal ‘pet’ ownership, and can convey a false sense of acceptability of exploiting animals for “cheap titillation”. Indeed, this has been reflected in a recent research review conducted on animal–visitor interactions in zoos from a number of different countries and global regions. These are unintended consequences that ”modern, ethical zoos” would try to minimise, or avoid completely where possible, though most zoos still offer close-contact experiences with their animals. Three ethical frameworks that may be beneficial for ethically run zoos to incorporate when considering human–animal interactions are: Compassionate Conservation, Conservation Welfare and Duty of Care. These three ethical frameworks are concerned with the welfare state and outcomes for individual animals, not just the population or species. Human–animal interactions in zoos may be acceptable in many circumstances and may be beneficial to both animal and human participants; however, they must be closely monitored through welfare tracking tools. The World Association of Zoos and Aquariums (WAZA) has published guidelines for human–animal interactions that are mandatory for member institutions to adhere to, although whether these guidelines are taken as mandatory or suggestions at individual institutions is unknown. Some suggestions for relevant extensions to the guidelines are suggested herein. Melding Duty of Care and the two Conservation ethical frameworks would be ideal for assessing the ethical acceptability of such interactions as they currently occur, and for considering how they should be modified to occur (or not) into the future in zoological settings.

## 1. Introduction

Human–animal Interactions (HAIs) are common occurrences in zoological institutions, from husbandry practices to interactions with visitors (both regulated and unregulated) [1,2]. Animal–visitor Interactions (AVIs) are often a large component of zoos’ appeal to visitors, and these experiences are also a large component of zoos’ operations and financial viability [1,3,4]. It has been estimated that global zoo attendance is over 700 million visitors annually [5]. Some of these zoo visitors attend purely for entertainment, and/or for direct interactions with animals (for which they are willing to pay) [4,6,7]; however, many visitors to modern zoos report considering zoos and aquaria as centres for education [8,9,10,11]. AVIs may be classified as “direct contact” (such as holding, feeding, brushing or touching experiences) or “indirect contact” (such as visually viewing, gaze-following and/or mimicking through shared enclosure windows, “scattering” food for the animals from a unique vantage point, auditory communication from traditional enclosure perimeters, or the “solving” of combined human–animal input “puzzle walls” installed in some zoo exhibits for “cognitive enrichment” of the enclosure animals). Globally, zoos vary significantly in their offering of direct and indirect contact animal experiences, but almost all zoos surveyed in a 2019 study promoted one or more types of interaction experiences on their public websites [4]. Yet, these interactions may be at odds with many of the ethical principles upon which “modern, ethical zoos” have built their new moral foundations, and expound their virtues and “social license” [1,4,12,13,14], such as ensuring positive welfare of their captive animals, promoting "natural behaviours", and being compassionate towards individuals as well as populations in their conservation efforts. This article discusses how three prominent ethical frameworks (which are often explicitly or implicitly utilised by zoos) may be used to examine and justify HAIs in zoos (examining interactions with both visitors and with zookeepers), how new guidelines for AVIs published by the World Association of Zoos and Aquariums (WAZA) [15] perform under these ethical frameworks, and whether the guidelines work in practice alongside the stated missions of some zoological associations and institutions. The three specific ethical frameworks discussed herein are: Compassionate Conservation, Conservation Welfare, and Duty of Care. These three frameworks are not mutually exclusive, although it is suggested here that a deliberate melding of elements and tenets from all three frameworks could make a robust new framework that would be of relevance to zoological institutions. Furthermore, these three frameworks are concerned with the welfare of individual animals, rather than whole populations or ecosystems as most other Conservation or Environmental ethical frameworks are. Many forms of Environmental ethics and Conservation ethics have been espoused over almost the last 100 years [16], with the collective aim of saving Earth’s last remaining wild and natural places from being paved over by human expansion/exploitation. These ethical frameworks are mostly characterised by a focus on the overall ecosystem health rather than on individual welfare outcomes [16,17]. These ethics have more recently been criticised for perpetuating the status quo of ecosystem or population health always trumping considerations of individual animals’ welfare [16] (and a lack of empathy for suffering individuals), for sidestepping problematic issues arising from our increasing knowledge of animal consciousness and sentience (and increasing knowledge of harmful anthropogenic impacts) [17], and for perpetuating the influential, anthropocentric “land ethic” attitude that species conservation is important, yet often only prioritised after human interests (especially where that land, or the animals on it, are of utility or economic benefit to humans) [18,19].

D’Cruze et al. [4] list five inter-connected goals that many modern zoos and aquaria share: 1. Conservation; 2. Education; 3. Research; 4. Animal welfare; and 5. Entertainment. While many modern facilities place major emphasis only on the first four goals, and shy away from promoting their facilities as places for human entertainment, as mentioned above, many visitors still report entertainment or leisure as their first reason for attending these places [4,6,7]. Many zoos and aquaria exist as private, for-profit enterprises, meaning a certain level of revenue is required to remain operational, and then profit is required to financially contribute to their conservation goals. WAZA report conservation as zoos’ core *purpose*, but their core *activity* is animal welfare [12]. Likewise, the American Association of Zoos and Aquariums (AZA) list their mission as “*helping member institutions and animals in their care thrive, through advancing animal welfare, public engagement, and the conservation of wildlife*” [20]; and the Australasian Zoo and Aquarium Association (ZAA) list saving (conserving) wildlife by inspiring best practice in conservation and (animal) welfare with support from government and community as their strategic mission for member institutions [21]. Both of these associations detail supporting member institutions’ financial and operational goals as key goals, as well as supporting and facilitating memorable visitor experiences, but they do not list “entertainment” as a key consideration in their strategic documents [20,21]. In fact, most accredited facilities oppose procuring and displaying animals for entertainment purposes, or training animals for “performances”, as part of their new “ecocentric” ethos [1,14]. It is important to note, too, that member institutions pay monetary fees and dues to continue to be members of these self-regulated associations, but the accreditation processes are independent of institutional membership. Accreditation processes with these associations are a benchmarking tool, for monitoring animal welfare standards and meaningful contributions to conservation within individual institutions [22,23]. Whilst human–animal interactions are not discouraged or banned by these associations, strict guidelines and policies around the acceptability of offering these (especially direct) interactions in accredited facilities are being written into modern documentation [13]. Here, the ethics and the welfare impacts of two types of HAIs shall be discussed: Animal–visitor Interactions; and lesser scrutinised Keeper–Animal Interactions (KAIs) and Relationships (KARs).

## 2. Human–Animal Interactions

HAIs have been extensively studied in the agricultural/production animal sector [24,25] and the effects of stockperson attitudes and behaviours on the behaviours and productivity of livestock have been well established, and typified in robust models, such as the Hemsworth–Coleman model [25] based on the psychological theories of reasoned action and planned behaviour [26,27]. Built upon the Hemsworth–Coleman livestock model, there are also a few models of HAIs in zoos, such as the Hosey model [28,29], and the Chiew–Hemsworth model of animal–visitor interactions (published in [30]). HAI research in zoos has steadily increased over the last few decades [14]. The results of many studies report mixed welfare effects of human interactions, from negative effects through to neutral and positive effects [4,14,31,32,33,34,35,36,37,38], and many of the results have been found to be very individual specific. Most studies of zoo HAIs to date have focused on assessing AVIs, and, so far, very few studies have assessed and quantified KAIs or KARs [32].

## 3. Animal–Visitor Interactions

There are now quite a few studies that have uncovered negative effects of visitor presence and interactions on captive zoo animal behaviour and welfare, especially when those interactions are in uncontrolled circumstances [4,34,37,38,39,40,41,42,43,44]. There are also many assumed detrimental (but currently unknown) effects of controlled interactions, such as in provided and promoted animal–visitor “experiences” within zoos, especially with understudied animals such as reptiles (e.g., handled snakes and lizards) [4,35,36,38]. Often, the current standards of housing conditions for these animals are also inadequate, however, and this is likely to increase or confound detrimental effects of other interactions or welfare-impacting conditions [45,46,47]. Although, there are also a number of studies that show that many zoo species are apparently unaffected by visitors and their behaviours, if only viewed from a distance (i.e., no direct physical interactions), and it has been supposed that these animals simply view visitors as a type of expected “environmental variation” [32,43,44,48,49,50].

Studies on the positive effects of AVIs are sparse [4], and are limited to very few species, such as lemurs [51,52], giant tortoises [35,53], and leopard tortoises [36]; and possible positive effects of visitors for orangutans [54,55] and meerkats [49]. Despite the dearth of research on positive AVIs, it is suggested here that, as research increases, more positive effects for some individual animals within captive groups (and possibly in some whole groups or populations) will be uncovered. This may further increase as AVIs are undertaken in a more controlled, ethical, and evidence-based manner, prioritising consideration of what the *animal* wants from the interaction, rather than the human [56]. When considering individual animal welfare as the ultimate priority for modern, ethical zoos [1,12] (especially those zoos adhering to Compassionate Conservation, Conservation Welfare and/or Duty of Care ethical frameworks), fostering positive AVIs (and positive HAIs in general) is of the utmost importance. There are countless anecdotal stories, passed between zookeeping and animal care staff, that exemplify positive human–animal interactions with animals under their charge. Properly recording and quantifying these relationships, to provide empirical evidence that these relationships are beneficial (or that they are not, in some circumstances) is suggested to be a next step in better understanding captive animals’ wants for, or against, these interactions.

## 4. Ethical Frameworks

### 4.1. Compassionate Conservation

Compassionate Conservation is an ethical framework that has flourished in the last decade, originally conceived to deal with many “wicked problems” [1] for individual animal welfare in wildlife management, that traditional Environmental and Conservation ethics could not effectively grapple with [16,57]. This framework has become an explicit ethical alignment within the code of ethics of some zoos, such as Zoos Victoria [1,58], although the framework as used in a pro-zoo manner [1] differs from the original Compassionate Conservation approach [57], which was largely concerned with wildlife management, and was generally aligned to anti-captivity principles. While its beginnings were of an in situ wildlife conservation focus, the principles of Compassionate Conservation as applied to ex situ conservation efforts within a captive zoo environment are largely the same [1]. Compassionate Conservation, in its different iterations, has been described by various proponents as ascribing mostly to a virtue ethic (the virtue of Compassion), a deontological ethic (Animal Rights theories), or to consequentialist ethics (the greatest good for the most number of animals) [59]. It is obviously a pluralistic approach, focused on the wellbeing of individual wild animals as well as larger populations and ecosystems. The main four tenets of Compassionate Conservation are: 1. First do no harm; 2. Individuals matter; 3. Inclusivity; and 4. Peaceful co-existence (an explanation of these principles is available in [59]). However, these tenets have also been criticised for a lack of clarity on how the specifics of this ethical framework can be applied to novel or complex dilemmas, such as individual suffering for the benefit of populations or ecosystems [59].

### 4.2. Conservation Welfare

A new ethic, Conservation Welfare (predicated mostly upon principles of Singer’s Utilitarianism [60]), has been proposed as a more legitimate and pragmatic framework for zoos, aquariums and other captive animal conservation organisations to become adherents of [59]. Conservation Welfare is the recent application of Animal Welfare ethics (and some principles of Conservation and Environmental ethics) to conservation practices for non-captive wild animals [59]. Like Compassionate Conservation, it differs from most Environmental ethics as it is largely focused on the wellbeing of individual animals, not just whole populations, species or ecosystems. Conservation Welfare, like Animal Rights and Compassionate Conservation, asserts that animals do indeed possess inherent value, meaning they are morally relevant, though the difference in Conservation Welfare is that this inherent value does not preclude the possibility of the imposition of individual suffering or death, if it is necessary and for the “greatest good” (i.e., it can be “traded-off”). Still, this ethic always endeavours to minimise pain and suffering in individual animals. Thus, as applied to in situ and ex situ conservation practices, a Conservation Welfare ethic is more pragmatic than Compassionate Conservation, in that the direct imposition of *some* suffering on *some* individuals is deemed acceptable (and this will not violate any tenets) as long as this suffering is *necessary* and *justified*. Although, what is deemed necessary, justified suffering is still somewhat ambiguous [59].

### 4.3. Duty of Care

The Duty of Care ethical framework (which was initially a humanistic ethical framework for humans caring for humans, then companion animals, and then other domesticated animals) is often an implicitly nurtured approach within zoos, distributed amongst the new generation of animal care managers and husbandry staff, as this ethical framework also promotes a duty to provide positive welfare conditions to captive animals which aligns with modern zoos’ goals. That is, as guardians of captive animals, we have a moral duty to provide all levels of care to those animals [61,62], including the provision of opportunities for animals to have “*a life worth living*” or to be able to thrive in captivity [1,63,64]. The duty of care ethic is a reasonable melding of two ethics—a deontological “duty-based” ethic (a moral obligation towards another), and a “virtue-based” care ethic (both active provision of care to others, and internally “caring about” (i.e., consideration for) others) [61,62]. Duty of care as a concept reaches far beyond simply an ethical framework, with “*currency in legal, philosophical, ethical, and general animal protection discourse*” [62]. As opposed to Conservation and Environmental ethics at large, these three specific ethical frameworks above are all concerned with the welfare of individual animals rather than populations.

## 5. WAZA Guidelines for AVIs

WAZA have specifically published a set of “*Animal-Visitor Interaction Guidelines*” [15], based on their 2003 Code of Ethics [65] and their 2015 Animal Welfare Strategy [12]. There are six key recommendations for AVI’s listed in the document, with further subsections devoted to recommended procedures to meet these guidelines. The six recommendations are:
1.*Avoid having animals in any interactive experience that would compromise their welfare*.2.*Animals involved in direct contact situations should receive appropriate training for visitor interactions in order to reduce potential discomfort or stress responses*.3.*Make no unnecessary demands on animals and assure that visitors do not provoke or create discomfort or stress responses in the animals*.4.*Provide animals with choice of whether to participate or not in the interactions.Allow adequate rest time and assure that an animal displaying any indication that it does not want to participate is immediately removed from the interactive experience*.5.*All walk-through habitats, touch pools and petting areas/touch paddocks where animals are in close proximity to visitors should be of a suitable size to provide for species-appropriate needs and have suitable refuge areas for the animals*.6.*Any feeding during an interaction must be regulated so it is consistent with the animal’s overall appropriate diet and health care. This food must not be the only access to food or the whole diet for the animal and the animal must have choice whether to accept this food*.

Prima facie, these guidelines are sensible and easily interpretable ways for reducing the negative impacts of AVIs on animals. However, individual institutional adherence to these “guidelines” in varying regions may be incomplete, inadequate, or altogether ignored (in favour of financial viability or human experience, for example). Likewise, the auditing of guideline adherence seems to be self-prompted by each individual institution, rather than by a broader regulatory body. Institutional adherence to WAZA and regional association guidelines is largely unknown, or at least reviews are held confidentially. Properly assessing these guidelines would also take an individualistic approach, whereas many zoo facilities often keep “encounter groups” consisting of multiple animals of the same species, and often assess their welfare collectively. Individual welfare assessments are becoming more common globally, especially with the development of specific welfare-monitoring tools (following the Five Domains model), such as WelfareTrak^®^ (Chicago Zoological Society, Chicago, IL, USA) [14,66]. Other issues include interpretation of specific guidelines. For example, guideline 3 states, “*make no unnecessary demands on animals*”, though, what exactly necessary or unnecessary demands during human interaction encounters are is ambiguous. One of the most important guidelines is number 4—“*provide animals with choice of whether to participate or not*”. Choice and control over their immediate situation are now known to be important for an animal’s overall wellbeing and agency, which can lead to positive welfare states, and these concepts are currently being taught to new generations of zookeepers and animal husbandry professionals as crucial provisions for captive animals wherever possible and pragmatic [14,67,68]. It is also suggested that it would be pertinent to add an additional guideline here around safe interaction practices, as follows: “7. *Only interactions with non-dangerous animals should be allowed and conducted, and if there is a reasonable chance of harm (even if minimal) to either the human or the animal participants, these interactive experiences should be terminated immediately.*” That is, direct physical contact “experiences” with large predatory animals, such as Tigers, Lions, Bears or Orcas, which could potentially cause serious injury or death to the human participants, should not be offered nor conducted by modern, ethical zoological and aquarium facilities. Currently, many of these offered experiences rely on harmful or abusive training practices, physical restraint, bodily mutilations (such as declawing or teeth removal), and punishments to maintain physical and psychological “control” over these large dangerous animals [69]. This does not preclude the possibility of beneficial positive HAIs between keepers and these animals, nor in fact between unfamiliar visitors and these animals, but direct contact in these situations is always of the highest risk. It should also be mentioned that most *accredited* zoological facilities have prohibited abusive and/or bodily mutilation practices in their codes of ethics [13,15,65], yet these practices still persist at many eco-tourism or unregulated destinations in many regions [69].

## 6. Keeper–Animal Interactions

Currently, AVIs are the focus of much research effort [4,66,70,71]. However, close examinations of keeper–animal interactions and relationships (KAIs; KARs) are sparse, with a few varying results [32,55,72,73,74,75]. Due to the persistence of many “*folklore husbandry*” practices [45], there is a strong possibility that we are currently ignoring many established negative relationships between zookeepers and animals under their charge [32]. Although, most modern zoological facilities and (nearly all) animal care professionals endeavour to minimise harmful interventions and to ameliorate possible negative HAIs *before* they become established negative HARs that would be detrimental to the animal’s overall welfare [1,75]. Furthermore, even though they are often communicated through folklore husbandry, many anecdotal stories and personal experiences (some documented in photographs or short videos) shared broadly over social media can sometimes be beneficial for improving KAIs and KARs in circumstances where objective, empirical evidence is not currently available. Folklore husbandry is a double-edged sword, however, and the established folklore is often very resistant to change even when presented with solid scientific evidence to the contrary [45,46].

To date, specific studies on positive KARs have found the following animal-focused results: increased reproductive success in small cats [76]; lower faecal glucocorticoid metabolites in clouded leopards [77], white rhinoceros [78], and Asiatic and African elephants [74]; reduced abnormal and stress-related behaviours after positive reinforcement training (PRT) in chimpanzees [79] and polar bears [80]; and increased responsiveness to husbandry cues after PRT in black rhinoceros, zebras and Sulawesi macaques [81]. Similarly, human-focused results found that zookeepers reported stronger, more positive KARs with tortoises that they conducted public-visible training sessions with [82]; another recent study found that zookeepers’ self-reported job dissatisfaction rose when “Keeper-Elephant Bonds” were weaker [74]. Apart from Alba et al. [82], all of these KAI studies have focused on mammalian species. Very little is known about KAIs with other classes of animal. It is strongly suggested that an increase in the empirical investigation of KAIs and KARs is necessary and warranted.

## 7. Are the Benefits Worth Allowing These Interactions?

As just described above, there are some reported benefits (for both humans and animals) of positive KARs in zoos. There is also marginal evidence to suggest that positive AVIs can be beneficial for the animals involved and documented evidence that these interactions do indeed improve visitor experiences, conservation caring and learning [4,70]. So, is there a good case for allowing and promoting AVIs in zoos? The answer is complicated, but yes. As with all complex dilemmas, the devil is in the details, as it were. Firstly, the guidelines as set out by WAZA, plus the suggested 7th recommendation above, should be closely adhered to, to prevent negative effects of close interactions. However, a new model for clearly identifying when these interactions are being “*asked for*” by captive animals needs to be developed (i.e., being more attentive to what animals actually “*want*”, and aware of how we interpret it [56]). Interactions that are “asked for” by animals means circumstances where animals have been observed “soliciting” interactions from people, either through glass or other barriers, or by direct contact at shared fence lines (as in the case of the Aldabran Giant Tortoises studied in [35]). Currently, many zoos have moved towards a highly “hands-off” model of animal keeping, such that most direct contact interactions between humans (both visitors and zookeepers) have been minimised, or totally abolished, and are discouraged as much as possible. Yet, this may be a counter to enhancing the overall welfare of animals in some circumstances, especially in situations where the animals are highly motivated to interact but are denied this rewarding outcome. Sufficient time should be dedicated by animal care managers to allow zookeepers or other staff qualified in animal behaviour to observe daily interaction solicitation or engagement by individual animals under their charge, to identify more opportunities for “*positive affective engagement*” interactions that may currently be overlooked or unnoticed. Furthermore, identifying specific individuals that may benefit from positive KAIs or AVIs should be prioritised by zoos as well. These animals may not always solicit interactions, but other personality factors may be apparent that could predict higher enjoyment of these interactions were they to be offered—factors such as high levels of boldness and curiosity are suggested to be a good starting point for investigation. For human participants, provision of these so-called “*profound experiences*” [1] in safe, controlled zoo environments can indeed be very beneficial for inciting pro-environmental and pro-conservation behaviour and attitudinal change in visitors, ultimately contributing to the zoos’ conservation goals in meaningful ways [3,43,66,83,84]. “Connecting” with wildlife has been rated as a top priority by zoo visitors, although the type of “connections” that they are seeking can vary significantly [71].

## 8. Unintended Consequences

Although AVIs may potentially be rewarding for all parties involved in some circumstances, there are also a number of risks associated with close contact experiences offered within zoos. Obviously, there are a number of health and safety issues for both animal and human participants that are involved in these interactions (especially direct physical contact interactions), some of which have been detailed elsewhere [4,14,69]. There is also a growing worry among zoo researchers, managers, educators and behaviour change specialists that providing opportunities to directly interact with animals in zoos may “normalise” the behaviours and promote a false sense of acceptability of engaging in these same behaviours in inappropriate circumstances, such as with wild animals or at unregulated “roadside zoos” and eco-tourism destinations with very poor animal welfare standards [4,69,85,86]. Interactive experiences that present these animals as “tame” or “cute” may also increase the desire to own these types of animals as exotic pets [87], and celebrities posing with animals at “roadside zoos” and poorly regulated eco-tourism destinations in social media posts can further normalise this problematic behaviour in unaware members of the public. There is a very real potential that “behavioural spill-over” [88] could occur after these experiences; thus, approach and interaction behaviours would be attempted by visitors in inappropriate circumstances (such as encounters with animals in the wild), especially because the interaction experienced in the zoo environment is likely to be highly rewarding emotionally and physiologically, leading to an increased motivation to engage in these types of behaviours more often [88].

Compounding these concerns, the ethical values and beliefs that certain individuals hold about interacting with wildlife may very likely increase the risks of inappropriate or ill-advised behaviours occurring. Historically, zoos were created as displays of imperial majesty—purely for elevating social/cultural status, human awe and entertainment [1,89,90,91]. Modern zoos are attempting to transform into ethical biodiversity conservation organisations that promote education and positive animal welfare [3,10,11,92,93,94,95], yet entertainment and leisure are still two commonly reported reasons for attending these destinations by patrons [1,89,90,91]. Indeed, a zoo visitor survey conducted by the author [96] found that one of the five extracted ethical alignments of visitors was labelled “*human interaction and entertainment priority*”. Visitors that aligned with this component had high agreement responses on questionnaire items such as “*humans should be allowed to interact with ALL animals in the zoo*”, “*zoo animals are like pets*”, “*zoo animals should be treated like pets*”, and “*I believe that it is acceptable to keep ALL types of animals in zoos*”. Patrons that hold these types of ethical views about interactions with wildlife are likely to be minimally concerned with the animal welfare risks associated with these interactions. They may also be less concerned with evaluating or acknowledging unsatisfactory animal handling and keeping conditions at unregulated, poor-welfare eco-tourism destinations, as their main priority in those moments is their own enjoyment (and they will engage in behaviours that are contrary to their usual moral attitudes) [97]. To counter this problem, engaging (yet stringent) educational elements must be built into interactive animal experiences offered by zoos, to attempt to change perceptions of these interactions as being harmless enjoyable interactions for all parties involved towards a realistic understanding of how the animals may actually feel about such interactions (and why this matters).

## 9. What Do the Ethical Frameworks Say?

From a Compassionate Conservation perspective, these types of human–animal interactions would usually be discouraged quite strongly. This is because there are many potential risks of harm to the animals involved (even though minor or non-existent in ideal settings), which would violate the first tenet. The repercussions and undesirable consequences listed above would also likely violate the tenet of peaceful co-existence, as most wild animals would be quite fearful or defensive towards humans approaching them for interactions. Whilst the controlled interactions in zoo environments could be beneficial to fostering pro-environmental attitudes if participants were educated correctly, the inherent risks of direct contact interactions are probably too great to allow. The Conservation Welfare framework would only allow these interactions to occur in very controlled circumstances, but would not completely discourage nor prohibit all of these types of direct interactions. The main principle that would have to be followed, however, is that only those interactions that are “asked” for by the captive animals (not the humans), and could be delivered in an absolutely safe and controlled manner, would be deemed acceptable. Although, uncontrolled HAIs at shared fence lines or through glass viewing windows would likely also be acceptable in circumstances where the animals were initiating or soliciting such interactions. Ergo, if the animal is “asking” for the interaction, and the interaction is deemed safe and minimal or zero risk, then this interaction could be used to increase both the individual animal’s wellbeing and welfare, and conservation caring in humans. Though Conservation Welfare would also be opposed to and concerned about negative “behavioural spill-over” into inappropriate circumstances with wildlife or poor welfare destinations, as this is counted-productive to conservation efforts and to fostering respect for nature. Duty of Care ethics would be mostly concerned with the impacts upon the individual animals within that particular captive environment, so many more HAIs in these circumstances would be deemed acceptable. The main principle followed would be to provide that which is best for the overall welfare for individual animals, and hence allowing and facilitating HAIs and AVIs that are positive and rewarding would be best practice. These interactions would have to be assessed for risks and for safety; however, the framework would only be concerned with the participants as they are in the immediate environment, not what the humans could potentially do in other circumstances or other times outside of the interaction. Therefore, effective communication and education for pro-environmental or conservation caring behavioural change in the human participants would not be considered a priority during these allowed interactions.

## 10. Conclusions: Promoting Positive Interactions

There are many potential risks inherent in HAIs in all circumstances. However, in specific settings there are also many potential benefits, with the potential to greatly enhance animal welfare conditions and human attitudes towards animal (and natural habitat) conservation and environmental caring. They could potentially be a very powerful tool to increase public awareness, engagement, and support for conservation practices and for achieving the goals of many zoological institutions. However, risks to animal and human participants, as well as the risks of inciting future inappropriate behaviours need to be thoroughly assessed and appropriately mitigated, and all direct HAIs should only be conducted in strictly “very low-risk” scenarios. There is great potential to vastly improve positive affective engagement in animals that are highly motivated to engage in these interactions, providing them with more choice and control over their captive environments [64,67,98]. Welfare monitoring tools (such as WelfareTrak^®^) should be utilised during all encounters, and direct behavioural observations from each and every session should be rigorously recorded, to ensure that only animals that are benefitting from interacting are continually used in “encounter programs”. Those animals that display fear, avoidance and/or defensive behaviours before, during, or after encounters should cease being used for these types of close-contact experiences. Behavioural observers must also become much more acutely aware of reptile species’ particular behaviours, as these animals’ full behavioural repertoires are still somewhat unknown [38]. Likewise, more accurate recognition of unreactive, torpid animals (that may be overwhelmed mentally and physiologically by both acute and chronic stressors) as animals that are not coping with their environments and/or handling must be treated as a priority for relevant behaviourists and animal care staff. Melding Duty of Care and the two Conservation ethical frameworks would be ideal for assessing the ethical acceptability of such interactions as they currently occur, and for considering how they should be modified to occur (or not) into the future in zoological settings.

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
