# Peer review of "Human–Animal Interactions in Zoos: What Can Compassionate Conservation, Conservation Welfare and Duty of Care Tell Us about the Ethics of Interacting, and Avoiding Unintended Consequences?"

_animals, 2020, doi:10.3390/ani10112037_

Round 1
Reviewer 1 Report
Overall this is an interesting commentary with some well thought out ideas. I particularly like the concept of merging ethical frameworks to provide an overarching method of evaluating AVIs in zoos. These are becoming more popular and common practice in zoos with more consideration (as outlined in this manuscript) needed for the ethical decisions around the use of animals in this way.
There are however a few issues with grammar throughout with some sentences running into each other and a need for appropriate punctuation in areas too. The author will also need to attend to the incorrect referencing on line 356.
Author Response
The author thanks the reviewer for their comments on the manuscript. Please find below a list of changes made.
REVIEWER: There are however a few issues with grammar throughout with some sentences running into each other and a need for appropriate punctuation in areas too. The author will also need to attend to the incorrect referencing on line 356.
AUTHOR REPLY
The author and another academic have proof read the manuscript again and have fixed any grammatical or typographic errors found.
The referencing on line 356 has been changed to correct MDPI format - now at line 386.
A list of other changes made to the manuscript by line:
L1: Sentence changed: “This article is an examination of human-animal interactions in zoos from an ethical perspective...”
L30-32: Sentence added: “The World Association of Zoos and Aquariums (WAZA) has published guidelines for human-animal interactions that are mandatory for member institutions to adhere to, although whether these guidelines are taken as mandatory or suggestions at individual institutions is unknown.”
L40-41: Sentence added: “Human-Animal Interactions (HAIs) are common occurrences in zoological institutions, from husbandry practices to interactions with visitors (both regulated and unregulated) [1,2].”
L56-62: Explanatory sentences added to introduction:
“This article discusses how three prominent ethical frameworks, which are often explicitly or implicitly utilised by zoos, may be used to examine and justify human-animal interactions in zoos (examining interactions with both visitors and with zookeepers), and how new guidelines for AVIs published by the World Association of Zoos and Aquariums (WAZA) [57] perform under these ethical frameworks and with the stated missions of some zoological associations and institutions. The three specific ethical frameworks discussed herein are: Compassionate Conservation, Conservation Welfare, and Duty of Care. These three frameworks are of relevance to zoological institutions as all three are concerned with the welfare of individual animals, rather than whole populations or ecosystems as most Conservation or Environmental ethical frameworks are.”
L67-76: New sentences added to introduce Conservation and Environmental ethics:
“Many forms of Environmental ethics and Conservation ethics have been espoused over almost the last 100 years [16]. These ethical frameworks are mostly characterised by a focus on overall ecosystem health rather than on individual welfare outcomes [16,17]. These ethics have more recently been criticised for perpetuating the status quo of ecosystem or population health always trumping consideration of individual welfare [16] (and a lack of empathy for suffering individuals); for sidestepping problematic issues arising from our increasing knowledge of animal consciousness and sentience (and increasing knowledge of harmful anthropogenic impacts) [17]; and for perpetuating the influential, anthropocentric "land ethic" attitude that species conservation is often prioritised after human interests (especially where that land, or the animals on it, are of utility or economic benefit to humans) [18,19].”
New references:
- Ramp, D.; & Bekoff, M. Compassion as a Practical and Evolved Ethic for Conservation. Bioscience, 2015, 65(3), 323. DOI: 10.1093/biosci/biu223446
- Fraser, D. A “practical” ethic for animals. J. Agr. Environ. Ethics,2012,25(5), 721—746.
- Leopold, A.A sand county almanac: And sketches here and there. Oxford University Press: Oxford, UK, 1949.
- Callicott, J.B. Whither Conservation Ethics? Conserv. Biol., 1990, 4(1), 15–20.
- Association of Zoos and Aquariums (AZA), North America. About AZA Accreditation, 2020. Available online: https://https://www.aza.org/what-is-accreditation (accessed on 26th October 2020).
- Zoo and Aquarium Association (ZAA), Australasia. ZAA Accreditation, 2020. Available online: https://www.zooaquarium.org.au/public/Public/Animal-Welfare/ZAA-Accreditation.aspx (accessed 26th433October 2020)
L56-58: Sentence amended: “… such as ensuring positive welfare of their captive animals, promoting "natural behaviours", and being compassionate towards individuals as well as populations in their conservation efforts.”
L146-149: New sentence: “Compassionate Conservation is an ethical framework that has flourished in the last decade, originally conceived to deal with wicked problems for individual animal welfare in wildlife management that traditional environmental and conservation ethics couldn’t effectively grapple with [16,57].”
L151-152: Sentence amended: “… which was largely concerned with wildlife management and was generally aligned to anti-captivity principles”.
L152-154: New sentence: “While its beginnings were of an in situ wildlife conservation focus, the principles of Compassionate Conservation as applied to ex situ conservation efforts within a captive zoo environment are largely the same [1].”
L155: Wording changed: “Compassionate Conservation, in its different iterations, has been described by various proponents as ascribing mostly to…”
Reviewer 2 Report
Line 1 “This article is an ethical examination of human-animal interactions in zoos” perhaps – this article is an examination of human-animal interactions in zoos from an ethical perspective (otherwise it reads as if the evaluation itself is ethical)
Lines 30-31 “published guidelines for human-animal interactions 31 that all member zoos must adhere to.” If they are guidelines can they also be mandatory – or are they mandatory for members only?
Introduction lines 37-79 – needs to say how the article is set out. This is important given comments below about “Ethical Frameworks”.
Lines 50-51 “ethical principles upon which…” what are these? If they are these points that follow in the next sentence “Conservation; 2. Education; 3. Research; 4. Animal 54 welfare;” – apart from animal welfare, how are these ethical principles?
Sentence on accreditation lines 72-74 is very broad and would benefit from and endnote reference.
Starting from line 121 Ethical frameworks - Make it clearer that the discussion is evaluating what ethical frameworks could be appropriate for zoos. I did not realise this until I got to line 134. Also given that from line 166 the material investigates “ WAZA Guidelines for AVIs” – it would be helpful for the author to make the reasons for this evaluation clearer. I gather that the reason for evaluation of ethical frameworks is to analyse which frameworks apply at an individual level and argue in favour of those (?) as has been done from line 316.
Regarding compassionate conservation – this needs clarification. The material eg refers to Zoos Victoria – however the mention of compassion in the document cited is different from compassionate conservation as espoused by Ramp/Bekoff etc. That movement largely deals with wildlife management (especially introduce, invasive, feral species) eg, Daniel Ramp and Marc Bekoff, “Compassion as a Practical and Evolved Ethic for Conservation”, (2015) 65 (3) Bioscience, 323; Daniel Ramp, Dror Ben-Ami, Keely Boom and David B Croft, “A Paradigm Shift for Wildlife Management in Australia”, in, Ignoring Nature No More: The Case for Compassionate Conservation, Marc Bekoff (ed), The University of Chicago Press, (2013) 295; Sara Dubois, Nicole Fenwick, Erin A. Ryan, Liv Baker, Sandra E. Baker, Ngaio J. Beausoleil, Scott Carter, Barbara Cartwright, Federico Costa, Chris Draper, John Griffin, Adam Grogan, Gregg Howald, Bidda Jones, Kate E. Littin, Amanda T. Lombard, David J. Mellor, “International Consensus Principles for Ethical Wildlife Control,” (2017) 31 (4) Conservation Biology, 753. Is this what the author is referring to? If so, how does it apply to zoos? If the author is referring to something different – needs to be explained. Lines 129-131 seem to pick up on the Ramp/Bekoff view of compassionate conservation – how does that apply to zoos?
Line 135 refers to Singer but is not referenced to him. It should also probably read “Singer’s Preference Utilitarianism” (?) This is important because Singer’s version of utilitarianism extends beyond simple aggregation of all pains and pleasures – although both are based on sentience.
Line 140 introduced conservation and environmental ethics out of the blue – need a better lead-in. Perhaps just one sentence to explain. This is especially important as lines 163-165 draw a conclusion that “As opposed to Conservation and Environmental ethics at 164 large, these three specific ethical frameworks above are all concerned with the welfare of individual 165 animals rather than populations.” – Hard for the reader to know whether they agree if, conservation/environmental ethics not explained briefly.
Line 141 “This ethic,” – probably best to say which one – lots of other ethics referred in between line 141 and the last mention of “Conservation Welfare”
Lines 166-243 – this is all very interesting and will be of value to readers.
Author Response
The author thanks the reviewer for their helpful and considered comments. The manuscript is certainly improved. Please find below a list of changes made by line, and responses to all comments.
REVIEWER
Line 1 “This article is an ethical examination of human-animal interactions in zoos” perhaps – this article is an examination of human-animal interactions in zoos from an ethical perspective (otherwise it reads as if the evaluation itself is ethical)
AUTHOR REPLY
Agreed. Change made.
L1: “This article is an examination of human-animal interactions in zoos from an ethical perspective”
REVIEWER
Lines 30-31 “published guidelines for human-animal interactions 31 that all member zoos must adhere to.” If they are guidelines can they also be mandatory – or are they mandatory for members only?
AUTHOR REPLY
They are considered mandatory for member institutions only, but whether they are taken as mandatory or simply a suggestive “guide” at individual institutions is unknown. Sentence re-worded to reflect uncertainty of adherence.
L30-32: “The World Association of Zoos and Aquariums (WAZA) has published guidelines for human-animal interactions that are mandatory for member institutions to adhere to, although whether these guidelines are taken as mandatory or suggestions at individual institutions is unknown.”
REVIEWER
Introduction lines 37-79 – needs to say how the article is set out. This is important given comments below about “Ethical Frameworks”.
AUTHOR REPLY
Agreed. New sentences added to introductory paragraph to address this.
L40-41: “Human-Animal Interactions (HAIs) are common occurrences in zoological institutions, from husbandry practices to interactions with visitors (both regulated and unregulated) [1,2].”
L56-62: “This article discusses how three prominent ethical frameworks, which are often explicitly or implicitly utilised by zoos, may be used to examine and justify human-animal interactions in zoos (examining interactions with both visitors and with zookeepers), and how new guidelines for AVIs published by the World Association of Zoos and Aquariums (WAZA) [57] perform under these ethical frameworks and with the stated missions of some zoological associations and institutions. The three specific ethical frameworks discussed herein are: Compassionate Conservation, Conservation Welfare, and Duty of Care. These three frameworks are of relevance to zoological institutions as all three are concerned with the welfare of individual animals, rather than whole populations or ecosystems as most Conservation or Environmental ethical frameworks are.”
L67-76: New sentences added to introduce Conservation and Environmental ethics: “Many forms of Environmental ethics and Conservation ethics have been espoused67over almost the last 100 years [16]. These ethical frameworks are mostly characterised by a focus on overall ecosystem health rather than on individual welfare outcomes [16,17]. These ethics have more recently been criticised for perpetuating the status quo of ecosystem or population health always trumping consideration of individual welfare [16] (and a lack of empathy for suffering individuals); for sidestepping problematic issues arising from our increasing knowledge of animal consciousness and sentience (and increasing knowledge of harmful anthropogenic impacts) [17]; and for perpetuating the influential, anthropocentric "land ethic" attitude that species conservation is often prioritised after human interests (especially where that land, or the animals on it, are of utility or economic benefit to humans) [18,19].”
New references:
- Ramp, D.; & Bekoff, M. Compassion as a Practical and Evolved Ethic for Conservation. Bioscience, 2015, 65(3), 323. DOI: 10.1093/biosci/biu223446
- Fraser, D. A “practical” ethic for animals. J. Agr. Environ. Ethics,2012,25(5), 721—746.
- Leopold, A.A sand county almanac: And sketches here and there. Oxford University Press: Oxford, UK, 1949.
- Callicott, J.B. Whither Conservation Ethics? Conserv. Biol., 1990, 4(1), 15–20.
REVIEWER
Lines 50-51 “ethical principles upon which…” what are these? If they are these points that follow in the next sentence “Conservation; 2. Education; 3. Research; 4. Animal 54 welfare;” – apart from animal welfare, how are these ethical principles?
AUTHOR REPLY
Added a few examples of ethical principles.
L56-58: “… such as ensuring positive welfare of their captive animals, promoting "natural behaviours", and being compassionate towards individuals as well as populations in their conservation efforts.”
REVIEWER
Sentence on accreditation lines 72-74 is very broad and would benefit from and endnote reference.
AUTHOR REPLY
References for accreditation added.
L92: “[22,23]”.
New references:
22. Association of Zoos and Aquariums (AZA), North America. About AZA Accreditation, 2020. Available online:430https://https://www.aza.org/what-is-accreditation (accessed on 26th October 2020).
23. Zoo and Aquarium Association (ZAA), Australasia. ZAA Accreditation, 2020. Available online: https://www.zooaquarium.org.au/public/Public/Animal-Welfare/ZAA-Accreditation.aspx (accessed 26th433October 2020).”
REVIEWER
Starting from line 121 Ethical frameworks - Make it clearer that the discussion is evaluating what ethical frameworks could be appropriate for zoos. I did not realise this until I got to line 134. Also given that from line 166 the material investigates “ WAZA Guidelines for AVIs” – it would be helpful for the author to make the reasons for this evaluation clearer. I gather that the reason for evaluation of ethical frameworks is to analyse which frameworks apply at an individual level and argue in favour of those (?) as has been done from line 316.
AUTHOR REPLY
Agreed. This has now been addressed in the introductory paragraph as above.
See lines 56-67.
REVIEWER
Regarding compassionate conservation – this needs clarification. The material eg refers to Zoos Victoria – however the mention of compassion in the document cited is different from compassionate conservation as espoused by Ramp/Bekoff etc. That movement largely deals with wildlife management (especially introduce, invasive, feral species) eg, Daniel Ramp and Marc Bekoff, “Compassion as a Practical and Evolved Ethic for Conservation”, (2015) 65 (3) Bioscience, 323; Daniel Ramp, Dror Ben-Ami, Keely Boom and David B Croft, “A Paradigm Shift for Wildlife Management in Australia”, in, Ignoring Nature No More: The Case for Compassionate Conservation, Marc Bekoff (ed), The University of Chicago Press, (2013) 295; Sara Dubois, Nicole Fenwick, Erin A. Ryan, Liv Baker, Sandra E. Baker, Ngaio J. Beausoleil, Scott Carter, Barbara Cartwright, Federico Costa, Chris Draper, John Griffin, Adam Grogan, Gregg Howald, Bidda Jones, Kate E. Littin, Amanda T. Lombard, David J. Mellor, “International Consensus Principles for Ethical Wildlife Control,” (2017) 31 (4) Conservation Biology, 753. Is this what the author is referring to? If so, how does it apply to zoos? If the author is referring to something different – needs to be explained. Lines 129-131 seem to pick up on the Ramp/Bekoff view of compassionate conservation – how does that apply to zoos?
AUTHOR REPLY
Some clarifying sentences have been added to address these issues.
L146-149: New sentence: “Compassionate Conservation is an ethical framework that has flourished in the last decade, originally conceived to deal with wicked problems for individual animal welfare in wildlife management that traditional environmental and conservation ethics couldn’t effectively grapple with [16,57].”
L151-152: Sentence amended: “… which was largely concerned with wildlife management and was generally aligned to anti-captivity principles”.
L152-154: New sentence: “While its beginnings were of an in situ wildlife conservation focus, the principles of Compassionate Conservation as applied to ex situ conservation efforts within a captive zoo environment are largely the same [1].”
L155: Wording changed: “Compassionate Conservation, in its different iterations, has been described by various proponents as ascribing mostly to…”
New reference: 16. Ramp, D.; & Bekoff, M. Compassion as a Practical and Evolved Ethic for Conservation. Bioscience, 2015, 65(3), 323. DOI: 10.1093/biosci/biu223
REVIEWER
Line 135 refers to Singer but is not referenced to him. It should also probably read “Singer’s Preference Utilitarianism” (?) This is important because Singer’s version of utilitarianism extends beyond simple aggregation of all pains and pleasures – although both are based on sentience.
AUTHOR REPLY
Added reference to Singer’s Animal Liberation, 2009 edition. “Singer, P. Animal Liberation: The Definitive Classic of the Animal Movement; Harper Perennial: New York, NY, USA, 2009. ISBN 9780061711305”
REVIEWER
Line 140 introduced conservation and environmental ethics out of the blue – need a better lead-in. Perhaps just one sentence to explain. This is especially important as lines 163-165 draw a conclusion that “As opposed to Conservation and Environmental ethics at 164 large, these three specific ethical frameworks above are all concerned with the welfare of individual 165 animals rather than populations.” – Hard for the reader to know whether they agree if, conservation/environmental ethics not explained briefly.
AUTHOR REPLY
Agreed. Sentences added in introduction on Conservation and Environmental ethics. See additions at lines 67-76 in comment above.
REVIEWER
Line 141 “This ethic,” – probably best to say which one – lots of other ethics referred in between line 141 and the last mention of “Conservation Welfare”
AUTHOR REPLY
Agreed. Changed.
L167: “This ethic” changed to “Conservation Welfare”.
REVIEWER
Lines 166-243 – this is all very interesting and will be of value to readers.
AUTHOR REPLY
The author thanks the reviewer very much for this comment.